# The Promising Effect of Tocilizumab on Chronic Antibody-Mediated Rejection (cAMR) of Kidney Transplant

**DOI:** 10.3390/pharmaceutics17010078

**Published:** 2025-01-09

**Authors:** Łukasz Świątek, Miłosz Miedziaszczyk, Dominik Lewandowski, Filip Robakowski, Piotr Tyburski, Marta Jakubowska, Marek Karczewski, Ilona Idasiak-Piechocka

**Affiliations:** 1Students Research Group of Transplantation and Kidney Diseases, Poznan University of Medical Sciences, 60-355 Poznan, Poland; luk.swi@outlook.com (Ł.Ś.);; 2Department of General and Transplant Surgery, Poznan University of Medical Sciences, 60-355 Poznan, Poland; 3Department of Clinical Pharmacy and Biopharmacy, Poznan University of Medical Sciences, 60-806 Poznan, Poland

**Keywords:** cAMR, DSA, eGFR, kidney rejection, tocilizumab

## Abstract

**Background**: Chronic antibody-mediated rejection (cAMR) constitutes a serious challenge in the long-term success of organ transplantation. It is associated with donor-specific antibodies (DSAs) which activate a complement pathway in response to the presence of human leukocyte antigens (HLAs) on the graft, which results in chronic inflammation and leads to graft dysfunction. One of the recent promising methods of cAMR treatment is a recombinant humanized anti-interleukin-6 receptor (IL-6R) monoclonal antibody referred to as Tocilizumab (TCZ). The aim of the presented systematic review is to explore the existing knowledge regarding the effect of tocilizumab treatment on cAMR and to perform a meta-analysis of the available data. **Methods**: A systematic review was performed using the PRISMA 2020 Checklist and Flow diagram. A systematic review protocol was registered in PROSPERO: CRD42024510996. The bias assessment was obtained with Methodical Index for Non-Randomized Studies (MINORS), whereas meta-analysis was performed using MedCalc. **Results**: Five clinical trials with a total number of 105 patients were included in our review. The mean loss of eGFR in time was −0.141 mL/min/1.73 m^2^ (95% CI: −0.409 to 0.126; *p* = 0.298) and was found to be statistically insignificant. The heterogeneity was low and was equal to I^2^ = 0.00%. The authors demonstrated a reduction in DSA titer by TCZ (−0.266 MFI (95% CI: −0.861 to 0.329; *p* = 0.377)). In the majority of studies, eGFR stabilization was associated with a reduction in DSAs. **Conclusions**: TCZ pharmacotherapy insignificantly reduced DSA titer and eGFR. Despite promising outcomes of potential eGFR stabilization, there is a need for large randomized controlled trials comparing standard management of cAMR and tocilizumab treatment.

## 1. Introduction

The number of kidney transplantations worldwide has been rising in the past years, reaching 24,273 procedures performed in the United States alone in 2019 [1]; owing to technological advances, this figure is expected to increase in the future. However, long-term survival remains low, mainly due to late allograft loss, which constitutes one of the major concerns of graft recipients. The resulting kidney rejection is classified according to the following types: hyperacute—which happens minutes following transplant, acute—which occurs within days to months after the procedure, and chronic—which develops up to three months to years after the transplantation [2].

The clinicopathological manifestation of chronic kidney rejection consists primarily of renal function deterioration in the initial 6–12 months post-transplant [3]. Graft dysfunction is diagnosed on the basis of commonly used laboratory tests, such as serum creatinine (sCr), blood urea nitrogen (BUN), and cystatin C. The estimated glomerular filtration rate (eGFR), is considered a relevant predictor of graft loss, reflecting the extent of structural and functional damage [4]. A 1-year graft survival for kidney transplants from deceased and living donors was observed in 94% and 97% of patients, respectively [5]. In fact, the Clinical Trials in Organ Transplantation consortium demonstrated a significant association between a 20–40% decline in eGFR within 3–24 months, or 6–24 months post-transplantation, and graft loss within 2–5 years [6]. Moreover, an increased risk of graft failure was associated with the production of post-transplant donor-specific antibodies and proteinuria [7,8]. Symptoms on physical examination in chronic kidney rejection are usually non-specific and include fatigue, fever, flu-like symptoms, anuria or decreased urine output, generalized edema, and pain or tenderness at the transplantation site [9]. It is of note that chronic antibody-mediated rejection (cAMR) represents a serious challenge in organ transplant and constitutes a significant risk for the long-term success of organ transplantation [10,11]. Recent studies have emphasized the role of donor-specific antibodies (DSAs) in cAMR pathogenesis. DSAs activate a complement pathway in response to the presence of human leukocyte antigens (HLAs) on the graft. This, in turn, leads to the formation of immune complexes, which contribute to chronic inflammation, fibrosis, and vascular remodeling, thus leading to graft dysfunction over time. Nevertheless, the exact composition and dynamics of immune complexes in cAMR may vary among individuals and depend on the specificities of DSAs and the HLA profile of the donor and recipient [4,12] Thus, understanding the intricate mechanisms of cAMR is crucial for developing effective diagnostic and therapeutic approaches. Current diagnostic tools, which involve advanced histopathological assessments, donor-specific antibody and C4d monitoring as well as molecular profiling, facilitate early detection and risk stratification. cAMR is diagnosed based on three criteria: morphologic evidence of chronic tissue injury (one or more of the following: severe peritubular capillary basement membrane multilayering, arterial intimal fibrosis of new onset); evidence of current/recent antibody interaction with vascular endothelium (including one or more of the following: linear C4d staining in peritubular capillaries, at least moderate microvascular inflammation); and serologic evidence of donor-specific antibodies (DSA to HLA or other antigens) [13,14]. The guidelines clearly distinguish between de novo donor-specific antibodies (DSAs) and pre-formed DSAs. Additionally, therapeutic objectives should focus on several factors, such as decreasing the rate of deterioration in glomerular filtration, proteinuria, histological injury score, and the titer of donor-specific antibodies, simultaneously minimizing drug toxicity [15].

The treatment of chronic antibody-mediated rejection remains complex. The majority of studies are characterized by small sample sizes, heterogeneity, and retrospective trial designs [16]. Moreover, there is a wide variety of treatment methods that can be introduced, which comprise plasma exchange, intravenous immunoglobulins, steroid pulse, anti-CD20 antibodies (rituximab, obinutuzumab), anti-IL-6 antibodies (tocilizumab, clazakizumab), anti-CD38 monoclonal antibody (daratumumab), proteasome inhibitors (bortezomib, carfilzomib), and complement-based therapy. In view of the diversity of available strategies, a comparison of their treatment effects poses a challenge [12,16].

One of the new promising methods of cAMR treatment is a recombinant humanized anti-interleukin-6 receptor (IL-6R) monoclonal antibody called Tocilizumab (TCZ) [17]. The main applications for this drug include systemic juvenile idiopathic arthritis (sJIA), polyarticular juvenile idiopathic arthritis (pJIA), and rheumatoid arthritis [RA] [18]. Since IL-6 plays a vital role in the differentiation of B-cells involved in antibody secretion and immunological response, it was suggested that targeting IL-6 by TCZ could be beneficial in patients presenting with cAMR [17]. Bearing in mind the potential positive outcome of the new treatment, the aim of this systematic review is to analyze the existing knowledge with regard to the impact of Tocilizumab on chronic antibody-mediated rejection of a kidney.

## 2. Materials and Methods

### 2.1. Search Strategy

The research strategy is based on search terms. The investigation was performed according to the PRISMA 2020 Checklist (Appendix A) and PRISMA 2020 Flow Diagram [19]. The authors employed PubMed and Cochrane search engines where they entered the following queries: ‘Tocilizumab’ AND ‘kidney chronic rejection’ OR ‘kidney humoral rejection’. The search included both randomized controlled trials and non-randomized controlled trials. Due to the small number of studies, the year of publication did not serve as a criterion. The review protocol was registered in the International Prospective Register of Systematic Reviews (PROSPERO registration ID: CRD42024510996).

### 2.2. Inclusion Criteria

Inclusion criteria were established on the basis of the PICO framework. The population studied consisted of patients with Chronic Active Antibody-mediated Rejection (cAMR) (Population; P). The intervention (I) involved treatment with Tocilizumab at a dosage of 8 mg/kg, up to a maximum of 800 mg. The expected outcomes were to be compared to baseline values (Comparison; C), with final outcomes assessed by comparing the initial and final estimated glomerular filtration rate (eGFR) at the study’s conclusion (Outcomes; O).

### 2.3. Exclusion Criteria

Exclusion criteria comprised follow-up periods of less than six months, studies with fewer than ten patients, and research published in languages other than English. Due to the small number of studies, no additional exclusion criteria were applied. The inclusion and exclusion criteria are summarized in Table 1.

### 2.4. Extracted Variables

Variables extracted from the studies included the year of publication, group size, initial and final eGFR, percentage of graft loss, and *p*-values. The overall outcome of the assessment is presented in Table 2. A secondary data search covered the immunological response parameters, including mean fluorescence intensity (MFI) or DSA level.

### 2.5. Data Collection Process

The selection process was conducted by three independent researchers. All disagreements were resolved through discussion, and the consensus statement is presented in the text below. During the identification stage, 42 records were obtained from PubMed databases. After eliminating duplicates, 38 records were subjected to screening, and out of these 26 were excluded for the following reasons: 16 due to inadequate form (review papers, case studies, case series), and 10 due to inadequate topic. The remaining 12 full papers underwent further scrutiny, and, finally, a total of 5 studies were included in the current review. The data collection process is shown in Figure 1 in the form of a Prisma Flow Diagram.

Notably, 7 papers that were initially deemed eligible for review were later excluded, since they either did not meet the inclusion criteria or fulfill the exclusion criteria. The studies conducted by Massat et al. [20] and Chamoun et al. [21] were excluded due to insufficient sample size (n1 = 9, n2 = 5, respectively)—these fulfilled the exclusion criterion. Another study which failed to meet the eligibility criteria was conducted by Arrive et al. [22], where its duration of 3 months was considered insufficient—this fulfilled the exclusion criterion. In turn, the paper by Shin et al. [23], describing the impact of tocilizumab on immunoglobulins and anti-HLA antibodies, lacked accurate eGFR data. Finally, a number of studies, such as those by Choi et al. [24] and Jordan et al. [25], failed to meet the inclusion criteria due to incomplete data. In fact, both trials provided insufficient information with regard to either initial or final eGFR. The complete dataset of extracted variables is presented in Appendix A.

### 2.6. Bias Assessment

Bias assessment was conducted using the Methodical Index for Non-Randomizes Studies (MINORS) [26]. The assessment was performed by two researchers, working independently. Written consensus was provided in the form of protocols for each individual study (Appendix A). In terms of quality, the following score ranges were established for non-comparative trials: 15–16—good quality, 9–14—moderate quality, ≤8—poor quality. Only papers which qualified as “moderate quality” and “good quality” were included in the final review. The evaluation outcome is demonstrated in Section 3.5. The presented review accounted for reporting bias, researchers worked independently, and the selection process for the studies did not involve country of origin, citation, or outcome.

**Table 2 pharmaceutics-17-00078-t002:** The effects of Tocilizumab on renal parameters in patients treated for chronic antibody-mediated rejection of kidneys.

Author, Year, Reference	Patients (n)	Initial eGFR [mL/min per 1.73 m^2^]	Final eGFR/Slope [mL/min per 1.73 m^2^]	Graft Loss [n] (%)	Follow-Up (Median)
Kumar et al., 2020 [27]	10	42 ± 19	39.2 ± 19 (at 6 months); 37 ± 24 (at the end)	2 (20%)	Median 12 months (8–24)
Lavacca et al., 2020 [28]	15	49.8 ± 13.4	48.4 ± 34.6	1 (6.7%)	Median 20.7 months
Noble et al., 2021 [17]	40	43 ± 17	41.6 ± 17	6 (15%)	12 months
Khairallah et al., 2023 [29]	29	41 ± 17	34 ± 15—after 3 months; 36 ± 15—after 6 months	2 (5.3%)	6 months
Boonpheng et al., 2023 [30]	11	57 ± 18	56 ± 17—after 6 months	0 (0%)	Median 12 months (3–18)

### 2.7. Statistical Analysis

Meta-analysis was performed using the Hedges g statistic as the formula for the standardized mean difference (SMD) and then the heterogeneity statistic was used to calculate the summary SMD in the random effects model. Egger’s test was used to assess publication bias, and the evaluation of heterogeneity was performed using the I^2^ statistic. Statistical analysis and creation of the attached figures were performed using MedCalc software, version 23.0.9. The outcome of the meta-analysis is presented in Figure 2, Figure 3, Figure 4 and Figure 5.

## 3. Results

### 3.1. Description of Studies

The studies selected for the purpose of the current review present an overall decrease in eGFR. Kumar et al. observed a non-significant change in eGFR over time, with mean eGFR 42 ± 18 mL/min per 1.73 m^2^ to 37 ± 24 mL/min per 1.73 m^2^; *p* = 0.27. The slope was also found to be insignificant (−0.14 ± 0.9 to −0.33 ± 1.1; *p* = 0.25) [27]. Lavacca et al. showed a reduction in eGFR over a period of 12 months following the introduction of TCZ (eGFR t0 vs. 12 months; 49.8 ± 13.4 vs. 48.4 ± 34.6 mL/min, *p* = 0.006) [28]. The research conducted by Noble et al. demonstrated no significant difference between the initial and final eGFR 41.6 ± 17 vs. 43 ± 17 mL/min/1.73 m^2^ (*p* = 0.102), which in turn suggests no significant change during a 12-month follow-up [17]. There was a significant difference in eGFR slopes between the pre-treatment period and the time after the initiation of TCZ. The slope difference was calculated to be 2.6 mL/min/1.73 m^2^ (SE = 0.8, *p* = 0.002). In their study, Khairallah et al. concluded that the introduction of TCZ not only reduced eGRF decline, but also reduced the inflammatory markers in patients with chronic active antibody-mediated rejection (CAAMR) [29]. According to Boonpheng et al., an insignificant change was observed after 6 months, from 57 ± 18 mL/min/1.73 m^2^ (range 28–89) of mean eGFR to 56 ± 17 mL/min/1.73 m^2^ (range 29–80) at 6 months (*p* = 0.25) [30].

### 3.2. Meta-Analysis

The studies containing sufficient data on the effect of TCZ on cAMR demonstrated a non-significant decrease in GFR in patients following kidney transplantation. The standardized mean difference in the random model was −0.141 mL/min/1.73 m^2^ (95% CI: −0.409 to 0.126; *p* = 0.298). The heterogeneity between studies was I^2^ = 0.00% (95% CI for I^2^ = 0.00 to 0.00; *p* = 0.9644). The results of Egger’s test—0.2910 (95% CI = −2.0938 to 2.6757; *p* = 0.7237), showed that there was a statistically non-significant publication load. The results are presented in Figure 2. The studies containing sufficient data on the effect of TCZ on cAMR demonstrated a non-significant decrease in DSAs in patients following kidney transplantation. The standardized mean difference in the random model was −0.266 MFI (95% CI: −0.861 to 0.329; *p* = 0.377). The heterogeneity between studies was I^2^ = 52.34% (95% CI for I^2^ = 0.00 to 86.30; *p* = 0.6583). The results of Egger’s test at −2.4818 (95% CI = −55.4747 to 50.5111; *p* = 0.6583) show that there is a non-significant publication load.

### 3.3. Immunological Response

The overall immunological response, including the DSAs, differs between the analyzed papers, since they employed different scales in order to measure the immunological response to the treatment. The research conducted by Kumar et al. reported no significant change in mean fluorescence intensity (MFI) 7272 ± 6698 to 6273 ± 8480 *p* = 0.63 after 12 months of TCZ treatment. One patient, initially with negative DSAs, developed low-grade class 2 DSAs at the end of the study [27]. In contrast, a significant change in MFI decrease was observed by Lavacca et al.: 22,600, 21,700–23,700 pre-TCZ and 18,200, 12,650–22,150 post-TCZ; *p* = 0.002, which indicates early serological improvement [28]. In the study conducted by Noble et al., median MFI was not evaluated; even though the measured IFTA decreased over time, i.e., in 9–12 months, the iIFTA score was 0.6 ± 0.6 vs. 1.1 ± 0.8 at baseline, and was found to be marginally insignificant (*p* = 0.053) [17]. The research conducted by Khairallah et al. presented an overall decrease in MFI by −102 (SE = 416, *p* = 0.8) per month, yet the difference between the slopes was not significant [29]. The observed differences are summarized in Table 3.

### 3.4. Studies Characteristics

In order to access the heterogeneity of the studies’ characteristics, a number of possible confounders were identified and are summarized in Table 4. The additional data may contribute to identifying the possible variations between the populations studied in various papers. The obtained data included the age when Tocilizumab therapy was initiated, Immunosuppression at TCZ initiation, previous cAMR therapy, treatment strategy, and eGFR interval assessment. The youngest study population was 43 years in the study by Noble et al. [17], whereas the oldest population was involved in the research conducted by Boonpheng et al. [30]. The predominant immunosuppression scheme was TAC/MFF/steroids (prednisone) in all studies except for the study by Kumar et al. [27], where 7/10 patients received belatacept instead of tacrolimus.

### 3.5. Bias Assessment

Bias assessment of the five studies was performed using the MINORS tool, and none of them were classified as “bad quality” in our analysis. The D5 domain (Unbiased assessment of the study endpoint) proved problematic in all the analyzed papers, since none of them included blinding in their data assessment. Another common issue was the D3 domain (Prospective collection of data), where the majority of studies failed to report data concerning the study protocol established before the onset of the study. Nevertheless, it is of note that Noble et al. obtained one point in the D3 domain, although the information provided regarding the protocol was insufficient [17]. The highest score was achieved by the research conducted by Lavacca et al. [28] with a total of 11 points, thus categorizing their study as a “moderate quality” paper. The second highest result (10 points) was achieved by three studies [17,27,29], which were also classified as “moderate quality”. The study by Boonpheng et al. [30] received the lowest score of 9 points, rendering their paper in the “moderate quality” category. However, as mentioned in the section devoted to methods, all five studies were ultimately included in the review. Bias assessment is summarized in Figure 6.

## 4. Discussion

### 4.1. Significance of the Changes in eGFR Values

According to our meta-analysis, the overall decrease in eGFR was −0.141 mL/min/1.73 m^2^ (95% CI: −0.409 to 0.126; *p* = 0.298), which was found to be insignificant and indicated there was no vital difference within the studied population following 6 months of treatment. Furthermore, it emphasized that no significant decrease in eGFR was observed in patients treated with TCZ. It is of note that the eGFR slope in patients with kidney rejection may reach up to −8.5 mL/min/1.73 m^2^ within 12 months (−11.4, −5.6) *p* < 0.001 prior to biopsy-proven diagnosis of late ABMR [31].

The randomized controlled trial comparing the 25 mg dose of clazakizumab and placebo injections in patients with AMR ≥ 365 days post-transplantation revealed a significantly slower decrease in eGFR in the clazakizumab group (−0.96; 95% CI, −1.96 to 0.03 vs. −2.43; 95% CI, −3.40 to −1.46 mL/min per 1.73 m^2^ per month) *p* = 0.04. This, in turn, indicates the clinical potential of other anti–IL-6 antibody drugs in the treatment of antibody-mediated rejection [32].

The decrease in eGFR is markedly associated with the risk of graft failure and death. The studies indicate that in patients with a eGFR decline of <−15 mL/min per 1.73 m^2^ between the first and third year, the death hazard ratio (HR) was equal to 1.77 (CI, 1.50 to 2.09) (c = 0.75), whereas individuals with ≥30% eGFR decline between the first and third year showed HR of 2.20 (CI, 1.87 to 2.60) (c = 0.75) [33].

### 4.2. Insight into the Standard Treatment

The treatment of cAMR remains a serious challenge, since no effective and approved treatment has been proposed up to date. Current recommendations for cAMR comprise plasmapheresis, IVIG, steroids, and rituximab; however, such management has not been FDA-approved [34]. Nevertheless, the main goals for almost all cAMR treatment strategies are to remove the circulating DSAs and to suppress the emergence of de novo donor-specific antibodies (dnDSA). Even though rituximab is commonly used for this purpose, the evidence supporting improved outcomes of rituximab treatment in kidney transplants with cAMR remains inconclusive [35,36,37]. Additionally, its administration is associated with an increased likelihood of severe infectious complications [37]. Some studies suggested that IVIG and steroids were associated with lower rates of graft rejection, although more research is necessary in order to fully determine their efficiency [16]. Moreover, according to other studies the development of cAMR is hypothetically linked to poor immunosuppression maintenance, due to the fact that immunosuppressive drugs (including prednisone, mycophenolate mofetil, and calcineurin inhibitors) are crucial in managing dsDNA levels [38]. Some researchers, in turn, recommend a more aggressive treatment (double filtration plasma-pheresis combined with one of the following: steroid pulses, rituximab, intravenous immunoglobulin, antithymogycte globulin, or bortezomib before advanced tissue injury occurs). In one of the retrospective analyses, a significant difference was observed in graft survival rate in favor of aggressive treatment compared to supportive therapy (routine treatment for chronic kidney disease) *p* = 0.015 [39]. One Korean study published in 2023 compared the effects of high-dose intravenous immunoglobulins (IVIG) with rituximab vs. a rituximab-only treatment. The findings showed no significant differences in terms of eGFR change (*p* = 0.988), allograft function, or proteinuria between the studied groups. In both groups, a significant decline was found in titer dnDSA at 12 months [40]. Another promising approach of cAMR treatment is the anti-body depletion strategy. In the study conducted by Shin et al., a significant reduction in IgG3 levels (1.5 ± 0.9 versus 0.8 ± 0.5 mg/mL, *p* = 0.007) was reported in patients treated with TCZ after 6 months, although the control group (IVIG + RTX) showed no significant changes in any of the IgG subclasses. In addition, the total levels of IgG, IgG1, IgG2 and IgG4 were found to be considerably decreased in the post-TCZ group [23]. Notably, the long-term therapeutic effect of TCZ was also confirmed in the study conducted by Choi et al. in 2017. They concluded that graft rejection and patient survival rate reached 80% and 91%, respectively, which was consistent with a positive treatment outcome. Moreover, DSAs started to decrease after 24 months, thus indicating the need for longer trials with regard to the administration of tocilizumab in cAMR [24].

### 4.3. Safety of Treatment

One of the most critical aspects of antibody-mediated rejection treatment is the risk of infectious complications. The 2021 research compared the prevalence of infections in patients treated with Tocilizumab at a dose of 8 mg/kg IV and IVIG 2 g/kg compared to those receiving rituximab at a dose of 375 mg/m^2^. The results demonstrated that fewer infections were observed in the TCZ group (463 infections/1000 patient-years vs. 730 infections/1000 patient-years in IVIG/rituximab group). The difference was found to be significant when *p* = 0.02, which indicated that TCZ treatment did not increase the risk of infection [41].

### 4.4. Limitations

This systematic review bears certain limitations. Only papers in English were included; translated manuscripts or studies published in other languages were excluded from our analysis. This limitation, i.e., a language bias, could affect information availability. Furthermore, since all the included studies were single-arm cohort trials and, therefore, there were no comparator groups, it constitutes a crucial element in the process of conclusion drawing by means of comparing the effects in two parallel groups. Additionally, some studies lost patients to follow-up, and because in certain cases such a loss exceeded 5%, a question emerges regarding the statistical significance of the included data.

## 5. Conclusions

Our meta-analysis emphasizes the promising potential of Tocilizumab in the management of chronic antibody-mediated rejection (cAMR). TCZ pharmacotherapy reduces DSA titer and does not lead to disruption in the filtration function of the transplanted kidney over a period of 6 months. Moreover, the observed low heterogeneity suggests a high degree of consistency in terms of the analyzed research findings. Nonetheless, it is essential to conduct further studies via randomized controlled trials, as the potential benefits both in terms of immunological and renal outcomes are significant, albeit currently corroborated by evidence of moderate quality.

## Figures and Tables

**Figure 1 pharmaceutics-17-00078-f001:**
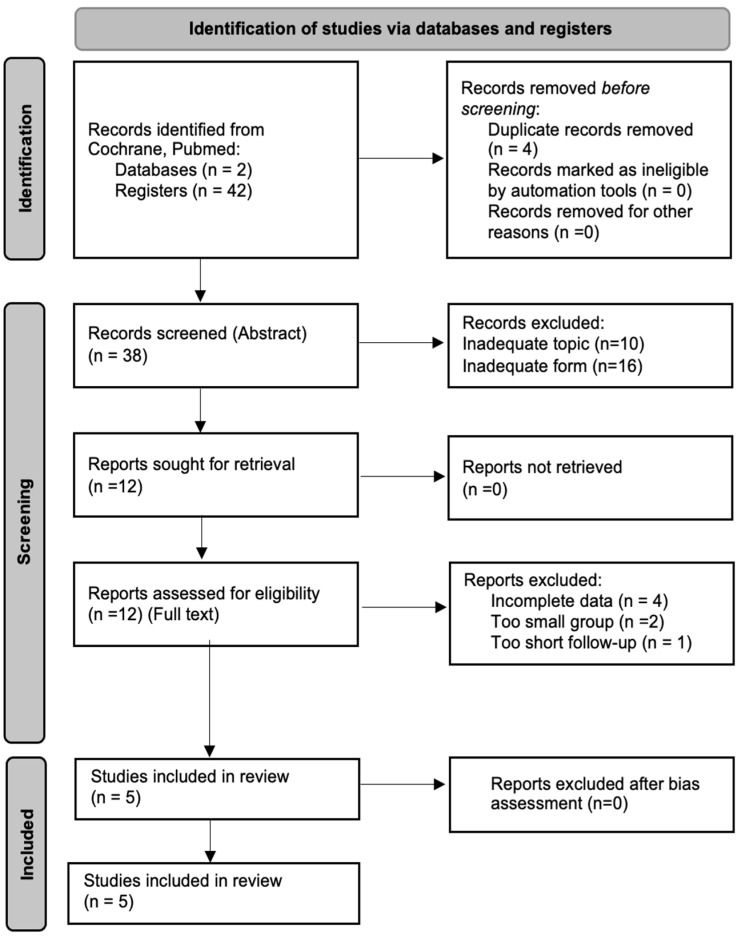
Data collection process—Prisma Flow Diagram.

**Figure 2 pharmaceutics-17-00078-f002:**
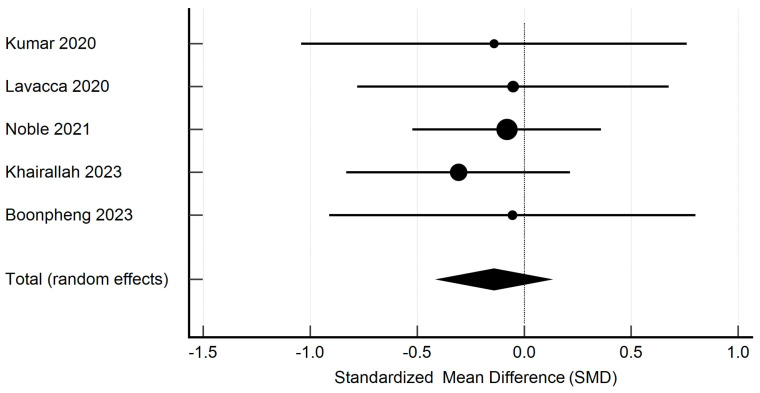
Forest plot presenting the results of our random effects meta-analysis for changes in eGFR [17,27,28,29,30].

**Figure 3 pharmaceutics-17-00078-f003:**
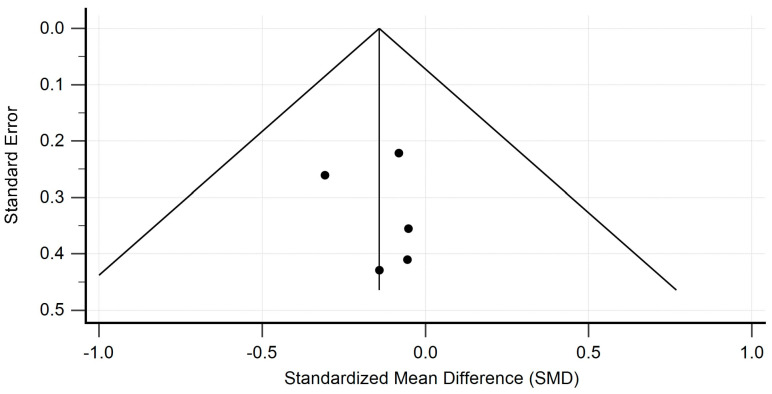
Heterogeneity funnel plot for eGFR.

**Figure 4 pharmaceutics-17-00078-f004:**
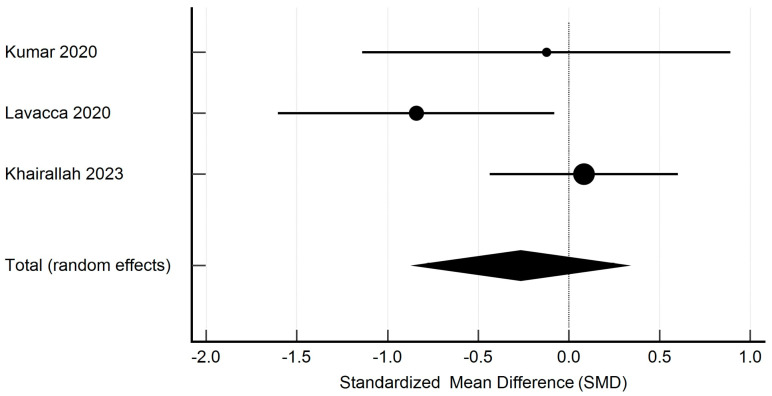
Forest plot demonstrating the results of our random effects meta-analysis in terms of changes in DSA titer [27,28,29].

**Figure 5 pharmaceutics-17-00078-f005:**
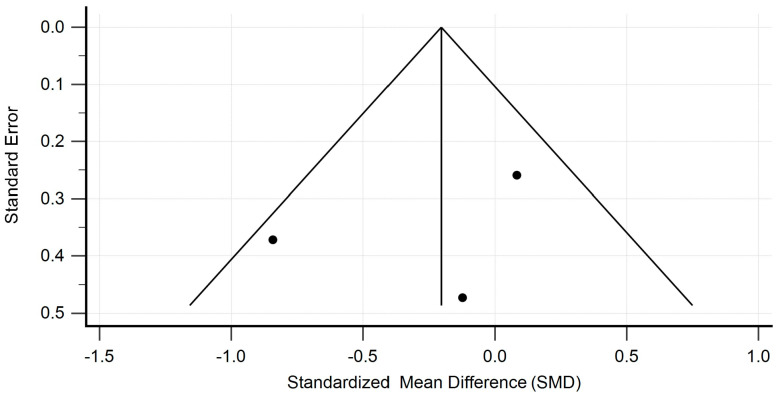
Heterogeneity funnel plot for DSA titer.

**Figure 6 pharmaceutics-17-00078-f006:**
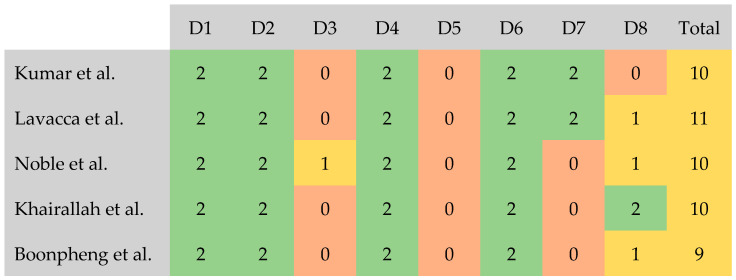
Bias assessment using MINORS [17,27,28,29,30]. D1—a clearly stated aim; D2—inclusion of consecutive patients; D3—prospective collection of data; D4—endpoints appropriate to the aim of the study; D5—unbiased assessment of the study endpoint; D6—follow-up period appropriate to the aim of the study; D7—loss to follow up less than 5%; D8—prospective calculation of the study size. Green color—2 points, Yellow—1 point, Red—zero points.

**Table 1 pharmaceutics-17-00078-t001:** The summary of the inclusion and exclusion criteria.

Inclusion Criteria	Exclusion Criteria
Patients diagnosed with cAMR (P) Patients treated with Tocilizumab 8 mg/kg, max 800 mg (I) No comparator, as only single-arm studies were available, initial and final values comparison (C) Comparison between initial and final eGFR (O)	Follow-up shorter than 6 months Studies involving fewer than 10 patients Non-English studies

**Table 3 pharmaceutics-17-00078-t003:** Immunological response in TCZ-treated patients.

Author, Year, Reference	Patients (N)	DSA+ Patients at TCZ Initiation	DSA1	DSA2	MFI Median at the Time TCZ Initiation (IQR)	Post-Tocilizumab MFI Median (IQR)	Follow up, (Months)
Kumar et al., 2020 [27]	10	8	1	7	7272 ± 6698	6273 ± 8480	12
Lavacca et al., 2020 [28]	15	15	1	14	22,600 (21,700–23,700)	18,200 (12,650–22,150) *	20.7
Noble et al., 2021 [17]	40	22	7	19	N/A	N/A	6
Khairallah et al., 2023 [29]	29	29	5	19	3450 (1350, 8875)	4000 (1600, 11,700)	6
Boonpheng et al., 2023 [30]	11	7	0	7	NA	NA	6

* Significant change *p* < 0.05. MFI—median fluorescence intensity, DSA1—donor-specific antibodies class 1. DSA2—donor-specific antibodies class 2.

**Table 4 pharmaceutics-17-00078-t004:** The treatment characteristics of analyzed studies included in the review.

Author, Year, Reference	Patients (N)	Median Age of (Years) TCZ Initiation	Immunosuppression at TCZ Initiation	Previous cAMR Therapy (% of Patients)	Treatment Strategy	eGFR Assessment Interval (Month)
Kumar et al., 2020 [27]	10	43.3	TAC 30% or BEL 70%, MMF, P	No	TCZ at the dose of 8 mg/kg per month.	0, 3, 6, 12
Lavacca et al., 2020 [28]	15	45.1	TAC, MMF, steroids 80%	No	TCZ at the dose of 8 mg/kg per month.	NA
Noble et al., 2021 [17]	40	43	TAC 85%, MMF 85%, mTOR 15%, BEL 15%, steroids 62%	82.5%	TCZ at the dose of 8 mg/kg per month.	0, 6, 12
Khairallah et al., 2023 [29]	29	41.2	TAC 92% or CsA 3% or BEL 3% or SIR 3%, MMF 92% or AZA 5%, P 55%	No	TCZ at the dose of 8 mg/kg per month In case of intolerance: 4 mg/kg	−3, 0, 3, 6
Boonpheng et al., 2023 [30]	11	52	TAC, MMF, P 91%	64%	TCZ at the dose of 8 mg/kg per month.	0, 6, 12

TAC—Tacrolimus, MMF—mycophenolate mofetil, P—prednisone, SIR—sirolimus, BEL—belatacept, AZA—azathioprine, mTOR—mTOR inhibitors.

## Data Availability

All research data are available upon reasonable request.

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
