# Peer review of "The Promising Effect of Tocilizumab on Chronic Antibody-Mediated Rejection (cAMR) of Kidney Transplant"

_pharmaceutics, 2025, doi:10.3390/pharmaceutics17010078_

Round 1

Reviewer 1 Report

Comments and Suggestions for Authors

The topic you chose and the systematic way you worked are to be appreciated. Yours inclusion and exclusion criteria were very well  specified.

It is a very well-designed review that followed the steps very well. However, in the end there are only 5 clinical studies included. It is a fairly small number and where the variability of the parameters chosen for comparison is high.

In the literature, I have found other Anti–IL-6 Antibody (Clazakizumab) involved in clinical trials for Late Antibody-Mediated Kidney Transplant Rejection. The conclusion of the study was: "Preliminary efficacy results suggest a potentially beneficial effect of clazakizumab and may therefore support the design of larger trials with a longer duration of follow-up."

Doberer K, Duerr M, Halloran PF, Eskandary F, Budde K, Regele H, Reeve J, Borski A, Kozakowski N, Reindl-Schwaighofer R, Waiser J, Lachmann N, Schranz S, Firbas C, Mühlbacher J, Gelbenegger G, Perkmann T, Wahrmann M, Kainz A, Ristl R, Halleck F, Bond G, Chong E, Jilma B, Böhmig GA. A Randomized Clinical Trial of Anti-IL-6 Antibody Clazakizumab in Late Antibody-Mediated Kidney Transplant Rejection. J Am Soc Nephrol. 2021 Mar;32(3):708-722. doi: 10.1681/ASN.2020071106. Epub 2020 Dec 18. PMID: 33443079; PMCID: PMC7920172.

Perhaps more data could have been obtained if more anti-interleukin-6 receptor (IL-6R) monoclonal antibody were included were included in the study, to see if there are indications in the management of chronic antibody-mediated rejection (cAMR).

Why didn't you include more anti-interleukin-6 receptor (IL-6R) monoclonal antibody?

The newest version of MedCalc is version 23.0.9. You use MedCalc software, version 20.006. Can you comment on this?

You mentioned using the Hedges g statistic with the fixed effects model, but I don't see the g values mentioned. Can you specify them?

Minor revisions

Paragraph 173 “The research conduct-ed” –

Paragraph 347 “papers were in-cluded”

Reviewer 2 Report

Comments and Suggestions for Authors

The authors undertook systematic review  to review the existing knowledge about effect of Tocilizumab on chronic antibody mediated rejection of kidney.  Although the study appears interesting, the following issues may be noted:

1.       The study has the inherent limitation of only 5 studies being selected for analysis. However, the key words used for conducting search lacks refinement, especially “kidney chronic rejection” and “kidney humoral rejection”. This will exclude many eligible studies. A random search on Pubmed with the above keywords give 1379 results , which is not mentioned in the manuscript. Please comment.

2.       The possible confounders needs to be identified. The details of treatment strategy adopted for the selected studies needs to be tabulated, as these can act as confounders in the improvement of eGFR.

3.       Did the authors looked into any secondary outcomes apart from eGFR?

4.       How do the authors account for the variability of follow-up periods in multiple studies ?

5.       What do the authors mean by “incomplete form” in case of rejection of articles. ?

Reviewer 3 Report

Comments and Suggestions for Authors

The systematic review aims to examine the existing knowledge on the effect of tocilizumab treatment on cAMR and to perform a meta-analysis of the extracted data.

The study provides sufficient information and has high scientific potential. The methods and results are adequately described and discussed.

Comments on the Quality of English Language

The English could be improved to more clearly express the research.

Reviewer 4 Report

Comments and Suggestions for Authors

I read the systematic review "The Promising Effect of Tocilizumab on Chronic Antibody Mediated Rejection (cAMR) of Kidney Transplant” with great interest.

The systematic review's methodology was well designed and sound and was applied in a very strict way, thus giving value to the generated data. I appreciated that the authors took time to explain in detail this methodology.

There are some issues that should be addressed. 

Lines 138-140: Two studies were presented as not meeting the inclusion criteria, in fact under 10 patients sample size being a exclusion criteria. So, basically the two studies presented a exclusion criteria. Please review and correct.

While one of the exclusion criteria was follow-up more than 6 months, in Table 2, for the Khairallah et al. study, median follow-up period was listed as "at least 3 months, 6 months following treatment”. Please explain better this values, because I presume there is a mistake somewhere.

Issues regarding the form of the text

In Table 1 - exclusion criteria - please add a bullet to the third exclusion criteria, “non-english studies”. In the current form, it may create confusions in readers, the third criteria appearing as a part of the second one.

Some words, are separated in syllables without being separated in different lines (probably they were in the initial manuscript; e.g. line 348 "consid-ered", line 300 “biop-sy-proven, line 253 “conduct-ed”etc. There are many such words. Please review and correct.

Round 2

Reviewer 2 Report

Comments and Suggestions for Authors

All comments have been addressed.

Author Response

Comment 1: 

All comments have been addressed.   Response 1 Authors want to thank for this comment, It is nice to see the appreciation of our work

Reviewer 4 Report

Comments and Suggestions for Authors

Thank you for answering the queries. In my opinion, there are still some technical issues to be fixed. Regarding the articles of Massat and Chamoun, they apparently meet all the inclusion criteria but were rightfully rejected because they also meet one exclusion criteria - small study size. These studies were listed in the paper among those rejected because they didn't meet inclusion criteria while, in reality, they were rejected because they had one exclusion criteria. I know that this is a strictly technical aspect but I strongly believe that the scientific papers must be as rigorous as possible, for best clarity while in front of the readers. Please review and correct.

Regarding the Khairallah study, the total number of studied patients was 38. If you focused only on the 6 months follow-up patient group (and all the calculations were based on these patients) then this should be clearly explained in the paper. Also, in the tables, you should include the number of only the patients followed for 6 months, instead of 38 (the entire patients group).

In Table 3, there is some text in Polish. Please review.
